# Added value of electrical impedance spectroscopy in adjunction of colposcopy: a prospective cohort study

Laura Bergqvist ![ORCID],[1] Annu Heinonen,[1] Xavier Carcopino,[2] Charles Redman,[3] Karoliina Aro,[1] Mari Kiviharju,[1] Seppo Virtanen,[1] Pirjo-Liisa Omar,[1] Laura Kotaniemi-Talonen,[4,5] Karolina Louvanto,[4,5] Pekka Nieminen,[1] Ilkka Kalliala[1,6]

PN and IK contributed equally.

For numbered affiliations see end of article.

**Correspondence to**
Dr Laura Bergqvist;
laura.bergqvist@hus.fi

## ABSTRACT

**Objective** To assess whether electrical impedance spectroscopy (EIS) as an adjunctive technology enhances the performance of colposcopy.

**Design** Prospective cohort study.

**Setting** University Hospital colposcopy clinic.

**Participants** Colposcopy with EIS for 647 women and conventional colposcopy for 962 women.

**Interventions** Comparison of the performance of colposcopy by referral cervical cytology in two cohorts, with and without EIS as an adjunctive technology.

**Outcome measures** Prevalence of cervical intraepithelial neoplasia grade 2 or worse (CIN2+), diagnostic testing accuracy to detect CIN2+ with and without EIS and their relative differences between cohorts.

**Results** The prevalence of CIN2+ varied between the cohorts according to referral cytology: 17.0% after abnormal squamous cells of unknown significance referral cytology in EIS cohort and 9.1% in the reference cohort, 16.5% and 18.9% after low-grade squamous intraepithelial lesion (LSIL), 44.3% and 58.2% after atypical squamous cells, cannot exclude high-grade squamous intraepithelial lesion (HSIL) (atypical squamous cells that cannot exclude HSIL), and 81.9% and 77.0% after HSIL cytology, respectively. Sensitivity to detect CIN2+ was higher in the EIS cohort, varying from 1.79 (95% CI 1.30 to 2.45) after LSIL referral cytology to 1.16 (95% CI 1.09 to 1.23) after HSIL referral cytology, with correspondingly lower specificity after any referral cytology.

**Conclusions** Colposcopy with EIS had overall higher sensitivity but lower specificity to detect CIN2+ than conventional colposcopy. CIN2+ prevalence rates were, however, not consistently higher in the EIS cohort, suggesting innate differences between the cohorts or truly lower detection rates of CIN2+ for EIS, highlighting the need for randomised controlled trials on the effectiveness of EIS.

## INTRODUCTION

During the next decades, the incidence and prevalence of high-grade cervical disease will decrease in the developed countries due to human papillomavirus (HPV) vaccination programmes[1 2] and transition to primary high-risk HPV (hrHPV)-DNA test-based

## STRENGTHS AND LIMITATIONS OF THIS STUDY

⇒ The intervention and reference cohorts were both collected within the daily patient flow at the same colposcopy clinic.

⇒ The reference cohort was collected between 2013 and 2017 (n=962) and the electrical impedance spectroscopy cohort between 2018 and 2021 (n=647).

⇒ The prevalence of cervical intraepithelial neoplasia grade 2 or worse (CIN2+) in both cohorts was based on the histopathological data obtained at the first visit.

⇒ Diagnostic testing accuracy was calculated for the detection of CIN2+ in both cohorts.

⇒ We estimated the added value of electrical impedance spectroscopy compared with conventional colposcopy within and between cohorts stratified according to the referral cytology.

screening.[3] Consequently, colposcopy will become more challenging due to resulting lower positive predictive value. Therefore, to detect those in need of treatment, it will be essential to correctly identify the high-grade lesions and take biopsies at representative locations. Also, reliable means to rule out high-grade lesions without excessive number of biopsies or frequently repeated tests or colposcopies are needed.

The sensitivity and specificity of colposcopy in identifying uterine cervical high-grade precancerous lesions have been previously reported to vary between 66%–80% and 63%–95%, respectively.[4–7] Furthermore, the probability of detecting a high-grade disease at colposcopy is affected by the referral cytology, being higher after high-grade squamous intraepithelial lesion (HSIL) cytology than after atypical squamous cells of undetermined significance (ASC-US) or low-grade squamous intraepithelial lesion (LSIL) cytology results.[8]

ZedScan (Zillico)[9] is a hand-held device using electrical impedance spectroscopy (EIS) in identifying cervical pathology.[10] It is designed to provide guidance to colposcopist in biopsy taking by indicating the most abnormal cervical tissue area.[10] ZedScan measures the electrical properties of the cervical epithelium to differentiate precancerous and cancerous tissue from normal epithelium.[10–12] The area with the most abnormal impedance is reported visually, aiding the colposcopist in targeting biopsies.

The sensitivity of colposcopy has been suggested to increase with the use of EIS[10 13–17] even in women with low probability of high-grade cervical disease and with minor colposcopic changes, as its use is independent of visual findings in colposcopy.[12 18 19] The developers of the technology have been involved in most of the published studies. In women with persistent hrHPV positivity without cervical cytological changes, EIS has detected additional cases of cervical intraepithelial neoplasia grade 2 or worse (CIN2+) compared with women without EIS examination.[18] The benefit of EIS seems to vary depending on the referral cervical cytology, being most useful in terms of finding extra cases of CIN2+ in women with low-grade referral cervical cytology.[13 14 16 17] National Institute for Health and Care Excellence guidelines recommend further research on EIS.[20]

Our objective was to assess, stratified according to referral cytology, whether EIS combined with colposcopy increases the diagnostic testing accuracy of CIN2+ compared with conventional colposcopy in women referred to colposcopy for abnormal cervical cytology.

## METHODS
### Participants
All women (n=1609) in this study were examined between 2013 and 2021 at the outpatient colposcopy clinic of Helsinki University Hospital for a new referral for abnormal cytology. We included women if their cervical transformation zone (TZ) was type 1 or 2 (TZ1–2) and the information on both colposcopic impression and histopathological results were available. Exclusion criteria were transformation zone type 3 (TZ3), previous history of cervical cancer or large loop excision of the transformation zone (LLETZ) and pregnancy. Women referred for persistent hrHPV positivity without cytological changes were excluded due to the lack of sufficient control cohort as hrHPV testing as a part of primary screening was implemented in Helsinki region only in 2019.

The EIS cohort consisted of 647 women with colposcopy and ZedScan examination successfully performed between September 2018 and August 2021. The cohort was collected prospectively with non-consecutive patient recruitment. Under the study period ZedScan equipment was available at the colposcopy and used at the decision of the individual colposcopist. EIS examinations were done according to the manufacturer's protocol and all colposcopists had adequate training prior to using the device.

If active bleeding during colposcopy occurred, the EIS procedure was omitted.

We could not directly compare the performance of colposcopy alone against colposcopy with ZedScan as an adjunctive tool using only the EIS cohort, as these two events were not truly independent of each other in the routine clinical setting applied here. Therefore, we used a previously collected prospective cohort of 962 patients examined with conventional colposcopy in the colposcopy clinic of Helsinki University Hospital, Finland, between 2013 and July 2017 as the reference cohort (ISRCTN10933736),[21] with all women fulfilling the inclusion criteria included. Only the primary colposcopy after referral and its histological results were included in both cohorts.

Abnormal cervical cytology results were categorised according to the Bethesda system as ASC-US or worse. Histological results were reported according to WHO 2003, 2013 and 2020 classification. The evaluation of histopathological specimens, biopsies and LLETZ cones, was done by the gynaecological histopathologists of Helsinki University Hospital. The most severe histological diagnosis of all biopsies or LLETZ was recorded.

### Clinical procedures
All participants had a colposcopic examination with the application of acetic acid to the cervix. Subsequently, participants in the EIS cohort underwent a ZedScan examination. ZedScan readings were made from 10 to 12 points clockwise around the cervix. On the ZedScan reading, red colour points out the area with the highest probability of high-grade disease, amber colour indicates possible high-grade areas and the absence of high-grade disease is indicated with green colour. In most women, 12 measurements cover well the junction area of the cervix. However, it might be possible that minor areas are omitted in case of a very large cervix. After routine measurements (10–12 around the cervix) in case of suspected presence of CIN2+ by ZedScan, a particular single point mode can be used to localise more carefully the most abnormal area to be biopsied. In the EIS cohort, cervical biopsy sites were determined by the colposcopist based on both ZedScan results and colposcopic impression. The most severe histological diagnosis of all biopsies was recorded.

Random biopsies were not routinely taken in either of the cohorts. Colposcopy examination in both cohorts was based on Finnish Current Care Guidelines.[22] Five per cent acetic acid and Lugol's iodine were available at the discretion of individual colposcopist to assess the abnormal cervical areas for biopsy. The colposcopic impression was recorded as high-grade, low-grade or normal. Immediate LLETZ at initial visit ('select and treat'-approach) was performed when evaluated necessary according to Finnish Current Care Guidelines: HSIL referral cytology with a colposcopic impression of CIN2+ entitled to perform LLETZ at the initial colposcopy with consent from the patient.[22] After cervical cytology with glandular atypia favouring neoplasia the Finnish Current

Care Guidelines recommends immediate LLETZ irrespective of the age of the referred woman.[22]

## Data analysis

We compared the prevalence of histologically confirmed CIN2+ lesions between the EIS and reference cohorts and calculated the sensitivity, specificity, positive and negative predictive values for colposcopy in both cohorts for the detection of CIN2+ lesions, both overall and stratified according to the referral cervical cytology. The positive test result for the EIS cohort was defined as suspected presence of CIN2+ either by ZedScan and/or via colposcopic inspection. The test result was negative if both the colposcopic impression and ZedScan agreed on low-grade lesion or normal cervical finding, that is, absence of CIN2+ lesion. In the reference cohort, the positive test result was defined as a colposcopic impression of CIN2+ while the negative test result was defined as the absence of changes suggesting CIN2+ lesions. The most advanced histopathological results of the biopsies or LLETZ specimens taken at the initial visit were used as a reference standard in both cohorts. Women without biopsies and with negative ZedScan result and normal colposcopic impression as well as low-grade referral were considered true negatives. Even though colposcopy and EIS examination were not truly independent tests in the setting used, we still performed a sensitivity analysis within the EIS cohort and separately assessed diagnostic testing accuracy of colposcopy and EIS in that cohort alone as well.

Risk ratio and risk difference were used to compare the sensitivity and specificity between the EIS and reference cohorts. The p values<0.05 were considered statistically significant. All statistical analyses were performed using Stata/SE V.15 (StataCorp, College Station Texas, USA) and all statistical tests used were two-sided.

## Patient and public involvement

Patients and/or the public were not involved in the design, conduct, reporting or dissemination of this research.

## RESULTS

There were 1027 eligible women with adequate colposcopy and ZedScan examination performed in the EIS cohort. Altogether 68 women with other referral reasons than abnormal cervical cytology, 215 women with follow-up colposcopy visits and 97 women with missing data were excluded. In total, 647 women with new colposcopy referrals of abnormal cytology were included in the analysis (online supplemental figure S1, table 1). Of all ZedScan procedures 75% were conducted by three individual colposcopists. The reference cohort included 1383 eligible women. Of these, 86 women were excluded due to other referral reasons than abnormal cervical cytology, 174 for having TZ3, 143 for missing relevant clinical data and 18 for pregnancy. As a result, a total of 962 women fulfilled the inclusion criteria (online supplemental figure S1, table 1).

At least one biopsy was taken or an imminent LLETZ made in 625 (96.6%) women in the EIS cohort and 952 (99.0%) in the reference cohort. Only one biopsy was taken from one quarter of women 165 (25.5%) in the EIS cohort and among 109 (11.3%) in the reference cohort, whereas 22 (3.4%) women in the EIS cohort and 10 (1.0%) in the reference cohort had no biopsy. The average number of biopsies was 1.8 if at least one biopsy was taken in the EIS cohort and 2.3 in the reference cohort (table 1).

Altogether 222 (34.3%) women in the EIS cohort had CIN2+, including 5 (0.8%) cervical carcinomas and 14 (2.2%) adenocarcinoma in situ cases. In the reference cohort 391 (40.6%) women had CIN2+, including 7 (0.7%) cervical carcinomas and 15 (1.6%) adenocarcinoma in situ cases (table 1). The prevalence of CIN2+ was higher in the reference cohort among those referred for LSIL or atypical squamous cells that cannot exclude HSIL (ASC-H) cytology, whereas the prevalence of CIN2+ was higher in the EIS cohort after ASC-US and HSIL referral cytology (table 2, online supplemental table S1).

In the EIS cohort the overall sensitivity to detect CIN2+ was 94% (95% CI 90% to 97%) with corresponding specificity of 34% (95% CI 29% to 39%) (table 2, online supplemental table S1). The sensitivity varied according to referral cytology, being the lowest, 77%, for LSIL cytology (95% CI 61% to 89%) and the highest for HSIL cytology with 100% sensitivity (95% CI 95% to 100%) (table 2, online supplemental table S1). The specificity was lowest for HSIL cytology, 6% (95% CI 0% to 29%), and highest for ASC-US, 47% (95% CI 36% to 59%). EIS missed three low-grade referral cases of CIN2+ identified by the colposcopist (two cases of CIN2 and one CIN3). Colposcopic impression was less than CIN2 in 43 CIN2+ cases that were detected by ZedScan. A total of 13 cases (5.9%) of CIN2+ were missed by both ZedScan and the colposcopist (biopsies were still taken due to suspicion of low-grade lesion), including 2 adenocarcinoma in situ cases and 11 high-grade lesions (9 CIN2 and 2 CIN3 cases).

In the reference cohort, the overall sensitivity to detect CIN2+ was 68% (95% CI 63% to 73%) with corresponding specificity of 84% (95% CI 81% to 87%) (table 2, online supplemental table S1). The sensitivity to detect CIN2+ by colposcopic impression of CIN2+ was the lowest after LSIL cytology, 43%, and the highest after HSIL cytology, 86% (figure 1, table 2, online supplemental table S1). Overall, the colposcopic impression was less than CIN2+ in 31.7% (124/391) of CIN2+ cases and biopsies were taken due to suspicion of a low-grade lesion. Results stratified according to TZ type, age and referral cytology are presented in online supplemental table S2. There was no obvious impact of age on specificity or sensitivity within different cytologies (online supplemental table S2).

Compared with the reference cohort, the sensitivity to detect CIN2+ was higher in the EIS cohort overall, with risk ratio of 1.38 (95% CI 1.28 to 1.49), and after LSIL, ASC-H and HSIL referral cervical cytologies (table 3, online supplemental table S3). TZ1

**Table 1** Characteristics of the electrical impedance spectroscopy (EIS) cohort and the reference cohort

| | EIS cohort | | Reference cohort | |
| --- | --- | --- | --- | --- |
| | n=647 | % | n=962 | % |
| Mean age | 35.7 | | 35.4 | |
| SD, range | 9.3 (20.3–76.4) | | 9.6 (19.2–67.8) | |
| Age | | | | |
| <30 years | 175 | 27.1 | 295 | 30.7 |
| 30–44 years | 366 | 56.6 | 495 | 51.5 |
| ≥45 years | 106 | 16.4 | 172 | 17.9 |
| | 647 | 100.0 | 962 | 100.0 |
| Referral cervical cytology stratified by age | | | | |
| ASC-US | | | | |
| <30 years | 28 | 4.3 | 43 | 4.5 |
| 30–44 years | 52 | 8.0 | 28 | 2.9 |
| ≥45 years | 14 | 2.2 | 28 | 2.9 |
| LSIL | | | | |
| <30 years | 39 | 6.0 | 79 | 8.2 |
| 30–44 years | 153 | 23.6 | 224 | 23.3 |
| ≥45 years | 44 | 6.8 | 78 | 8.1 |
| ASC-H | | | | |
| <30 years | 72 | 11.1 | 90 | 9.4 |
| 30–44 years | 90 | 13.9 | 120 | 12.5 |
| ≥45 years | 30 | 4.6 | 27 | 2.8 |
| HSIL | | | | |
| <30 years | 31 | 4.8 | 75 | 7.8 |
| 30–44 years | 54 | 8.3 | 102 | 10.6 |
| ≥45 years | 9 | 1.4 | 23 | 2.4 |
| AGC-NOS | | | | |
| <30 years | 5 | 0.8 | 5 | 0.5 |
| 30–44 years | 15 | 2.3 | 12 | 1.2 |
| ≥45 years | 8 | 1.2 | 11 | 1.1 |
| AGC-FN | | | | |
| <30 years | 0 | 0.0 | 3 | 0.3 |
| 30–44 years | 2 | 0.3 | 9 | 0.9 |
| ≥45 years | 1 | 0.2 | 5 | 0.5 |
| | 647 | 100.0 | 962 | 100.0 |
| TZ type | | | | |
| TZ type 1 | 446 | 68.9 | 620 | 64.4 |
| TZ type 2 | 201 | 31.1 | 342 | 35.6 |
| | 647 | 100.0 | 962 | 100.0 |
| Biopsies and LLETZ | | | | |
| No biopsy | 22 | 3.4 | 10 | 1.0 |
| 1 biopsy | 165 | 25.5 | 109 | 11.3 |
| 2 biopsies | 263 | 40.6 | 420 | 43.7 |
| 3 biopsies | 83 | 12.8 | 257 | 26.7 |
| 4 biopsies | 1 | 0.2 | 43 | 4.5 |
| 5 biopsies | 0 | 0.0 | 5 | 0.5 |

**Table 1** Continued

| | EIS cohort | | Reference cohort | |
|---|---|---|---|---|
| | n=647 | % | n=962 | % |
| LLETZ | 113 | 17.5 | 118 | 12.3 |
| | 647 | 100.0 | 962 | 100.0 |
| Histology | | | | |
| No biopsy | 22 | 3.4 | 10 | 1.0 |
| Normal histology | 222 | 34.3 | 247 | 25.7 |
| CIN1 (LSIL) | 181 | 28.0 | 312 | 32.4 |
| CIN2 (HSIL) | 95 | 14.7 | 210 | 21.8 |
| CIN3 (HSIL) | 107 | 16.5 | 154 | 16.0 |
| Glandular atypia | 1 | 0.2 | 7 | 0.7 |
| AIS | 14 | 2.2 | 15 | 1.6 |
| Adenocarcinoma | 3 | 0.5 | 3 | 0.3 |
| Sq. cell carcinoma | 2 | 0.3 | 4 | 0.4 |
| | 647 | 100.0 | 962 | 100.0 |

AGC-FN, atypical glandular cells that favour neoplasia; AGC-NOS, atypical glandular cells not otherwise specified; AIS, adenocarcinoma in situ; ASC-H, atypical squamous cells that cannot exclude HSIL; ASC-US, atypical squamous cells of undetermined significance; CIN, cervical intraepithelial neoplasia; EIS, electrical impedance spectroscopy; HSIL, high-grade squamous intraepithelial lesion; LLETZ, large loop excision of the transformation zone; LSIL, low-grade squamous intraepithelial lesion; sq. cell carcinoma, squamous cell carcinoma; TZ, transformation zone.

and taking two or more biopsies were associated with higher observed sensitivity (table 3, online supplemental table S3). Specificity was correspondingly lower in the EIS cohort overall as well as when stratified according to referral cytology (table 3, online supplemental table S3).

In the EIS cohort, colposcopic impression of high-grade disease (CIN2+) was present with EIS indicating the presence of CIN2+ in 73.4% of all histologically confirmed CIN2+ cases. In the sensitivity analysis within the EIS cohort, colposcopy alone was indicative for the presence of CIN2+ in 166 of 222 CIN2+ cases (74.8%) and ZedScan in 206 of 222 (92.8%) of CIN2+ cases, suggesting an additional 40 cases (24.1%) detected by ZedScan only. The additional cases increased the detection of CIN2+ from 30 to 44 in women with low-grade cytology and from 136 to 162 in women with high-grade cytology (figure 1). The sensitivity to detect CIN2+ by colposcopy alone according to referral cytology was otherwise similar between the cohorts, except for women with ASC-H cervical cytology the colposcopy alone in the EIS cohort seemed to detect more CIN2+ cases (p=0.02) (figure 1). Among colposcopists who performed colposcopies in both cohorts, the average number of biopsies by cytology were higher in all cytology groups in the reference cohort compared with the EIS cohort. The average number of biopsies varied between 1.7 and 2.3 in the EIS cohort and between 2.2 and 2.8 in the reference cohort (online supplemental table S4).

## DISCUSSION

We compared the performance of colposcopy in detecting CIN2+ according to referral cervical cytology with and without EIS as an adjunctive technology. Colposcopy combined with EIS seemed to have a higher sensitivity, but a lower specificity compared with conventional colposcopy, regardless of the referral cervical cytology. The prevalence of CIN2+ lesions was higher in the EIS cohort after ASC-US and HSIL referral, but lower after LSIL and ASC-H cervical cytology. The average number of biopsies was lower in the EIS cohort.

Overall, EIS performed well with a high sensitivity (94%) but had a low specificity (34%) consistent with the previous studies.[13 14 16] Here, the sensitivity might have been overestimated in both cohorts as the true positive result was based on histology data at first visit only and lesions missed at first visit and detected during the follow-up were not included in either cohort. Still, this would not affect the estimates of relative performance. The sensitivity (68%) and specificity (84%) of colposcopy in the reference cohort were as well in line with existing data.[5 7 23]

The increased detection of CIN2+ cases by EIS has been reported as most pronounced in women with low-grade cytology[13 14 16 17] or with hrHPV positivity without cytological changes.[16 18] In our study, additional cases of CIN2+ detected by EIS were also most frequent among low-grade referrals. Furthermore, the sensitivity to detect CIN2+ with EIS was higher in most cervical cytology groups (ASC-US, LSIL, ASC-H, HSIL) compared with colposcopy alone. Only within HSIL cytology EIS

Table 2  Sensitivity and specificity of the electrical impedance spectroscopy (EIS) cohort and the reference cohort for the detection of CIN2+ lesions by cervical cytology, TZ type and age group

| | EIS cohort (n=647) | | | | Reference cohort (n=962) | | | |
|---|---|---|---|---|---|---|---|---|
| | CIN2+/n | Colpo+ZS CIN2+* | Sensitivity | Specificity | CIN2+/n | Colpo CIN2+† | Sensitivity | Specificity |
| All | 222/647 | 209 | 94 (90–97) | 34 (29–39) | 391/962 | 267 | 68 (63–73) | 84 (81–87) |
| ASC-US | 16/94 | 15 | 94 (70–100) | 47 (36–59) | 9/99 | 5 | 56 (21–86) | 97 (91–99) |
| LSIL | 39/236 | 30 | 77 (61–89) | 42 (35–49) | 72/381 | 31 | 43 (31–55) | 92 (89–95) |
| ASC-H | 85/192 | 84 | 99 (94–100) | 11 (6–19) | 138/237 | 87 | 63 (54–71) | 65 (54–74) |
| HSIL | 77/94 | 77 | 100 (95–100) | 6 (0–29) | 154/200 | 133 | 86 (80–91) | 46 (31–61) |
| AGC-NOS | 3/28 | 2 | 67 (9–99) | 44 (24–65) | 5/28 | 3 | 60 (15–95) | 96 (78–100) |
| AGC-FN | 2/3 | 1 | 50 (1–99) | 100 (3–100) | 13/17 | 8 | 62 (32–86) | 25 (1–81) |
| TZ1 | 156/446 | 146 | 94 (89–97) | 31 (26–37) | 279/620 | 187 | 67 (61–73) | 84 (80–88) |
| TZ2 | 66/201 | 63 | 95 (87–99) | 40 (32–49) | 112/342 | 80 | 71 (62–80) | 84 (79–88) |
| <30 years | 60/175 | 56 | 93 (84–98) | 35 (26–44) | 134/295 | 96 | 72 (63–79) | 77 (70–83) |
| 30–44 years | 131/366 | 124 | 95 (89–98) | 33 (27–40) | 211/495 | 144 | 68 (62–75) | 86 (81–90) |
| ≥45 years | 31/106 | 29 | 94 (79–99) | 35 (24–47) | 46/172 | 27 | 59 (43–73) | 89 (82–94) |
| HG cytology | 164/289 | 162 | 99 (96–100) | 11 (6–18) | 305/454 | 228 | 75 (70–80) | 58 (49–66) |
| LG cytology | 58/358 | 47 | 81(69–90) | 43 (38–49) | 86/508 | 39 | 45 (35–57) | 93 (91–96) |
| ASC-H, HSIL | 162/286 | 161 | 99 (97–100) | 10 (6–17) | 292/437 | 220 | 75 (70–80) | 59 (50–67) |
| ASC-US, LSIL | 55/330 | 45 | 82 (69–91) | 43 (37–49) | 81/480 | 36 | 44 (33–56) | 93 (90–96) |
| Glandular | 5/31 | 3 | 60 (15–95) | 46 (27–67) | 18/45 | 11 | 61 (36–83) | 85 (66–96) |
| 1 biopsy | 11/165 | 7 | 64 (31–89) | 51(43–59) | 14/109 | 5 | 36 (13–65) | 99 (94–100) |
| 2 biopsies | 78/263 | 70 | 90 (81–96) | 23 (17–30) | 112/420 | 66 | 59 (49–68) | 90 (86–93) |
| ≥3 biopsies | 43/84 | 43 | 100 | 0 | 168/305 | 113 | 67 (60–74) | 67 (59–75) |
| LLETZ | 90/113 | 89 | 99 (94–100) | 4 (0–22) | 97/118 | 83 | 86 (77–92) | 38 (18–62) |

*Colposcopic impression and/or ZedScan result of CIN2+ of histologically confirmed CIN2+ cases.
†Colposcopic impression of CIN2+ of histologically confirmed CIN2+ cases.
AGC-FN, atypical glandular cells that favour neoplasia; AGC-NOS, atypical glandular cells not otherwise specified; ASC-H, atypical squamous cells that cannot exclude HSIL; ASC-US, atypical squamous cells of undetermined significance; CIN, cervical intraepithelial neoplasia; EIS, electrical impedance spectroscopy; HG, high grade; HSIL, high-grade squamous intraepithelial lesion; LG, low grade; LLETZ, large loop excision of the transformation zone; LSIL, low-grade squamous intraepithelial lesion; TZ, transformation zone.

combined with colposcopy detected all CIN2+ cases. In women with other referral cytology (ASC-US, LSIL, ASC-H) there were cases of CIN2+ that EIS did not detect, but where biopsy of CIN2+ was warranted based on colposcopic diagnosis. Nevertheless, missed cases of CIN2+ were even more frequent in the reference cohort, where more CIN2+ lesions were detected in biopsies with colposcopic impression of CIN1 or lower. Contrary to expectations, the prevalence of CIN2+ was higher in the EIS cohort only after ASC-US and HSIL referral cytology. One explanation for lower prevalence of CIN2+ lesions in the EIS cohort after LSIL and ASC-H cytology could be that routine practice in Finland is to take biopsies also from low-grade lesions, rather than to abstain from taking biopsies when CIN2+ lesions are not colposcopically suspected. Biopsies even from mild acetowhite lesions are important in excluding a high-grade disease as the sensitivity of colposcopy to detect CIN2+ is far from 100%. Such biopsies could well have been more frequent without than with EIS as an additional confirmation on

suspected absence of CIN2+. This is supported by the observation that two or more biopsies were taken from 54% of women in the EIS cohort, whereas up to 75% of women in the reference cohort had at least two biopsies. In addition, the average number of biopsies by cytology among colposcopists who performed colposcopies in both cohorts were constantly higher in the reference cohort compared with the EIS cohort reflecting a change in manner/threshold to take biopsies when ZedScan was used as an adjunct technology. Multiple biopsies are known to increase the sensitivity of colposcopy as at least small lesions can easily be missed.[24 25] In women with low-grade referral cervical cytology, a single biopsy has shown to be insufficient to rule out a high-grade disease.[26] A British survey has also reported experienced colposcopists to take mostly two biopsies in diagnosing high-grade disease.[27] A Danish study found taking four biopsies to increase the detection rate of cervical dysplasia to 95.2%.[28] The average number of biopsies in the EIS cohort was higher (1.84) compared with previous reports

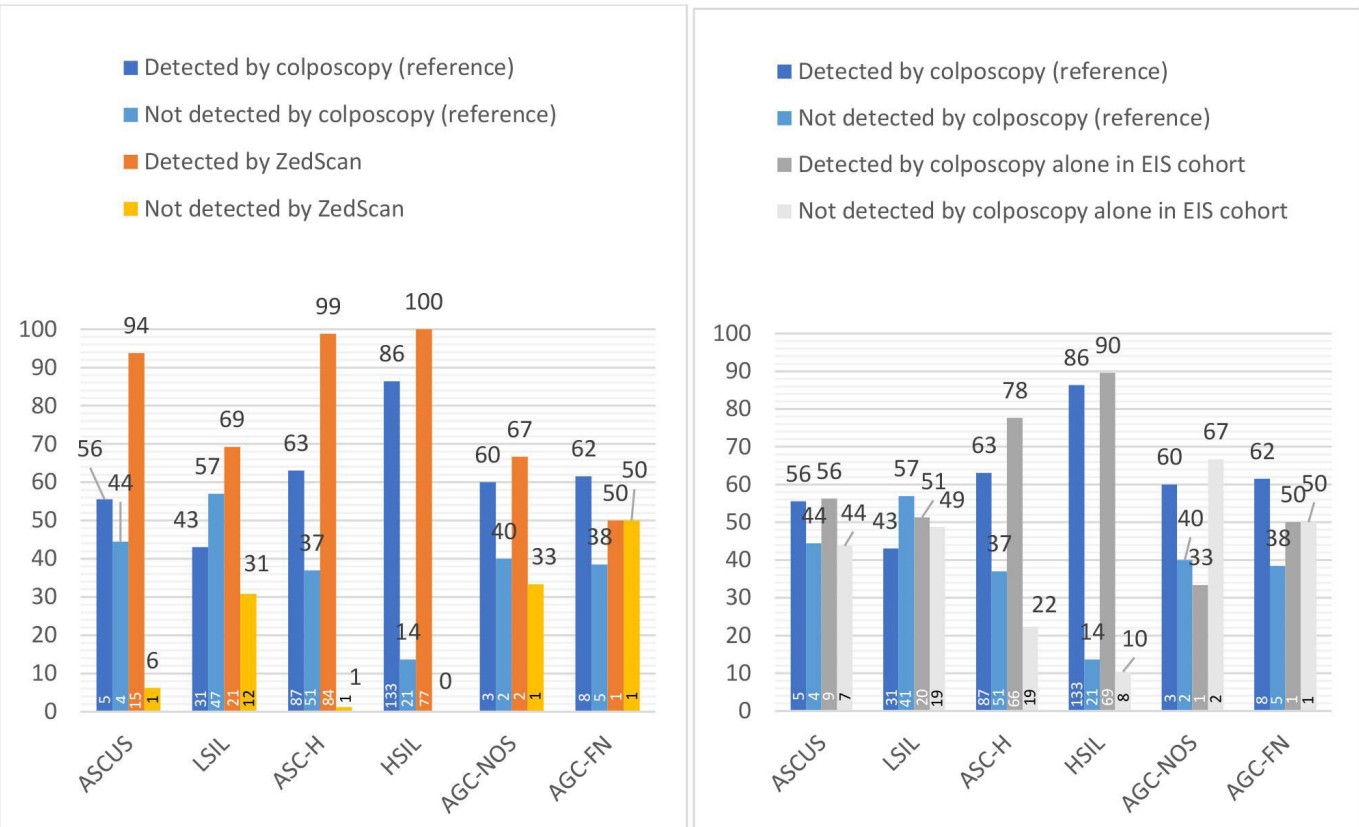

**Figure 1** Numbers and rates of CIN2+ lesions detected in the electrical impedance spectroscopy cohort (EIS) and in the reference cohort according to referral cytology. (A) Numbers and rates of CIN2+ detected by ZedScan alone and reference cohort stratified according to referral cytology. (B) Numbers and rates of CIN2+ detected by colposcopy alone in EIS and reference cohorts stratified according to referral cytology. Numbers of patients are given in the columns. AGC-FN, atypical glandular cells that favour neoplasia; AGC-NOS, atypical glandular cells not otherwise specified; ASC-H, atypical squamous cells that cannot exclude HSIL; ASC-US, atypical squamous cells of undetermined significance; CIN2+, cervical intraepithelial neoplasia grade 2 or worse; HSIL, high-grade squamous intraepithelial lesion; LSIL, low-grade squamous intraepithelial lesion.

(1.07 and 1.51),[13 14] but still lower than in the reference cohort (2.3).

Our observation of overall fewer biopsies along with fewer CIN2+ lesions detected in the EIS cohort can either indicate a true difference in CIN2+ prevalence between the cohorts, selection bias towards using EIS preferably on patients in whom CIN2+ lesion is not clearly present, or that CIN2+ lesions could have been missed in the EIS cohort, especially after LSIL and ASC-H referral cytology. If lesions were missed, it could possibly be due to a higher biopsy threshold in the EIS cohort, as indicated by lower number of biopsies. Without longitudinal data we still cannot be certain whether prevalent CIN2+ cases were indeed more frequently missed at the first visit in the EIS cohort. The prevalence of CIN2+ in EIS cohort in women with high-grade cytology (ASC-H and HSIL) is below previous observations (56.7% vs 79.1–84.0%).[13 16] However, when restricted to only women with HSIL referral cervical cytology or low-grade (ASC-US and LSIL) cytology, the prevalence for CIN2+ here did not differ from previous reports.[13 16] Cytological diagnoses may well vary between cytopathologists as well as between countries and this

possible difference in classification might also explain the observed difference in CIN2+ prevalence, especially after ASC-H cytology.[29] The longitudinal data on EIS results are scarce. In women referred with low-grade cytology, the future risk of CIN2+ was increased in up to 36 months follow-up if both colposcopic impression and EIS results were indicative for CIN2+ compared with women with other combinations of these two parameters, suggesting that EIS might provide new information on the future risk of high-grade disease.[30]

### Strengths and limitations
Most previous studies have compared the performance of EIS as an adjunctive technology for colposcopy against conventional colposcopy within the cohort where EIS was used, even though in clinical setting EIS is not a truly independent measurement from colposcopy. To our knowledge this is the first report on the performance of EIS as an adjunctive technology for colposcopy stratified according to referral cytology and compared with an external reference cohort. Even though our cohorts were collected at different time periods, they both represent women in the same catchment

**Table 3** Sensitivity and specificity of the electrical impedance spectroscopy cohort (EIS) and the reference cohort by cytology, TZ type and age group in identifying CIN2+, with corresponding risk ratios (RR) of sensitivity and specificity

| | EIS Sensitivity | Reference Sensitivity | Sensitivity RR (95%)* | P value | EIS Specificity | Reference Specificity | Specificity RR (95%)* | P value |
|---|---|---|---|---|---|---|---|---|
| All | 94 (90–97) | 68 (63–73) | 1.38 (1.28 to 1.49) | <0.0001 | 34 (29–39) | 84 (81–87) | 0.40 (0.35 to 0.46) | <0.0001 |
| ASC-US | 94 (70–100) | 56 (21–86) | 1.69 (0.93 to 3.07) | 0.0219 | 47 (36–59) | 97 (91–99) | 0.49 (0.39 to 0.62) | <0.0001 |
| LSIL | 77 (61–89) | 43 (31–55) | 1.79 (1.30 to 2.45) | 0.0006 | 42 (35–49) | 92 (89–95) | 0.45 (0.38 to 0.53) | <0.0001 |
| ASC-H | 99 (94–100) | 63 (54–71) | 1.57 (1.38 to 1.78) | <0.0001 | 11 (6–19) | 65 (54–74) | 0.17 (0.10 to 0.30) | <0.0001 |
| HSIL | 100 (95–100) | 86 (80–91) | 1.16 (1.09 to 1.23) | 0.0007 | 6 (0–29) | 46 (31–61) | 0.13 (0.02 to 0.89) | 0.0033 |
| AGC-NOS | 67 (9–99) | 60 (15–95) | 1.11 (0.38 to 3.25) | 0.8504 | 44 (24–65) | 96 (78–100) | 0.46 (0.29 to 0.72) | 0.0001 |
| AGC-FN | 50 (1–99) | 62 (32–86) | 0.81 (0.19 to 3.47) | 0.7565 | 100 (3–100) | 25 (1–81) | 4.0 (0.73 to 21.84) | 0.1709 |
| TZ1 | 94 (89–97) | 67 (61–73) | 1.40 (1.27 to 1.53) | <0.0001 | 31 (26–37) | 84 (80–88) | 0.37 (0.31 to 0.44) | <0.0001 |
| TZ2 | 95 (87–99) | 71 (62–80) | 1.34 (1.18 to 1.52) | 0.0001 | 40 (32–49) | 84 (79–88) | 0.48 (0.38 to 0.59) | <0.0001 |
| <30 years | 93 (84–98) | 72 (63–79) | 1.30 (1.15 to 1.48) | 0.0007 | 35 (26–44) | 77 (70–83) | 0.45 (0.35 to 0.59) | <0.0001 |
| 30–44 years | 95 (89–98) | 68 (62–75) | 1.39 (1.25 to 1.53) | <0.0001 | 33 (27–40) | 86 (81–90) | 0.39 (0.32 to 0.47) | <0.0001 |
| ≥45 y | 94 (79–99) | 59 (43–73) | 1.59 (1.23 to 2.07) | 0.0008 | 35 (24–47) | 89 (82–94) | 0.39 (0.28 to 0.54) | <0.0001 |
| HG cytology | 99 (96–100) | 75 (70–80) | 1.32 (1.24 to 1.41) | <0.0001 | 11 (6–18) | 58 (49–66) | 0.19 (0.12 to 0.32) | <0.0001 |
| LG cytology | 81 (69–90) | 45 (35–57) | 1.79 (1.37 to 2.33) | <0.0001 | 43 (38–49) | 93 (91–96) | 0.46 (0.41 to 0.53) | <0.0001 |
| 1 biopsy | 64 (31–89) | 36 (13–65) | 1.78 (0.77 to 4.10) | 0.1654 | 51 (43–59) | 99 (94–100) | 0.51 (0.44 to 0.60) | <0.0001 |
| 2 biopsies | 90 (81–96) | 59 (49–68) | 1.52 (1.28 to 1.81) | <0.0001 | 23 (17–30) | 90 (86–93) | 0.26 (0.20 to 0.34) | <0.0001 |
| ≥3 biopsies | 100 | 67 (60–74) | 1.49 (1.34 to 1.65) | <0.0001 | 0 | 67 (59–75) | 0 | <0.0001 |

*The values of risk ratio >1 imply better/improved effect with ZedScan.
AGC-FN, atypical glandular cells that favour neoplasia; AGC-NOS, atypical glandular cells not otherwise specified; ASC-H, atypical squamous cells that cannot exclude HSIL; ASC-US, atypical squamous cells of undetermined significance; CIN, cervical intraepithelial neoplasia; EIS, electrical impedance spectroscopy; HG, high grade; HSIL, high-grade squamous intraepithelial lesion; LG, low grade; LSIL, low-grade squamous intraepithelial lesion; TZ, transformation zone.

area referred to colposcopy due to abnormal cervical cytology. All colposcopies were performed in the same clinic by experienced colposcopists. Furthermore, none of the authors of this work have financial conflicts of interest with the technology studied. Our study also has some limitations. It is not possible to rule out that there would not have been any variation in sensitivity or specificity between the cohorts in different time periods. EIS device is not truly independent of colposcopic skills and the colposcopic performance can vary depending on the colposcopist. Also, the referral cytology and the colposcopic impression are incorporated in the EIS analysis by ZedScan. In order to take into account the variation of colposcopic performance and reliance on EIS device we collected a large cohort representing routine work. Including colposcopic examinations by several different colposcopists represents a real-life situation which could be considered as a strength compared with studies where all colposcopies have been performed by a single colposcopist.

When the cervical TZ is not fully visible, TZ3, ZedScan technology cannot be reliably applied and the results are not applicable to this population. CIN2+ lesions could well have been missed in both cohorts since the results are based on data collected on the initial visit. EIS might miss some lesions that either could have been detected with lower biopsy threshold or where biopsy would not have been indicated even in conventional colposcopy. However, complete certainty of the histology would have required LLETZ for all participants which would not have been ethically just.

## Conclusions

Colposcopy with EIS has a higher sensitivity and a lower specificity in identifying CIN2+ compared with conventional colposcopy, irrespective of cervical cytology. EIS can, therefore, be assumed to be of clinical benefit in colposcopy, particularly in women with low-grade cervical cytology where the prevalence of CIN2+ is low. We also observed an overall lower prevalence of CIN2+ lesions in the EIS cohort compared with a reference cohort with conventional colposcopy. The performance of EIS as an adjunctive technology for colposcopy has not been previously compared by cytology to an external reference cohort. While the observation of lower CIN2+ rate could be explained by different CIN2+ prevalence between the cohorts or selection bias, the finding is important and warrants further research, especially along with the observed lower number of biopsies in the EIS cohort. Adjunctive technologies are likely to become increasingly appealing in colposcopy, as the prevalence of high-grade cervical lesions is declining. Randomised controlled trials comparing EIS with a conventional colposcopy, including women referred due to persistent HPV infection without cytological changes are warranted. Before such further evidence, firm

recommendations on applicability of EIS as an adjunctive technology for colposcopy cannot be made.

**Author affiliations**
[1]Department of Obstetrics and Gynaecology, University of Helsinki and Helsinki University Hospital, Helsinki, Finland
[2]Department of Obstetrics and Gynaecology, APHM, AMU, Marseille, France
[3]Department of Obstetrics and Gynaecology, University Hospital of North Midlands, Stoke-on-Trent, UK
[4]Department of Obstetrics and Gynaecology, Tampere University Hospital, Tampere, Finland
[5]Department of Obstetrics and Gynaecology, Faculty of Medecine and Health Technology, Tampere University, Tampere, Finland
[6]Department of Metabolism, Digestion and Reproduction and Department of Surgery and Cancer, Institute of Reproductive and Developmental Biology, Faculty of Medecine, Imperial College, London, UK

**Contributors** PN acts as guarantor and accepts full responsibility for the work and/or the conduct of the study, had access to the data, and controlled the decision to publish.PN and IK were responsible for the conceptualisation and design of the study as well as methodology. XC, KL and LK-T contributed to conceptualisation. LB performed the statistical analysis with the aid of IK. LB, PN, MK, P-LO, SV and AH were responsible for data collection. LB drafted the original manuscript and IK, PN, CR, XC, KL, LK-T, AH and KA participated in writing, reviewing and editing. All authors listed qualify for authorship and approved the final version of the paper.

**Funding** Open access funded by Helsinki University Library. Finnish State Research Funding, Academy of Finland and Suomen lääketieteen säätiö are funding this manuscript.

**Competing interests** None declared.

**Patient and public involvement** Patients and/or the public were not involved in the design, or conduct, or reporting, or dissemination plans of this research.

**Patient consent for publication** Not applicable.

**Ethics approval** This study and data collection on patients where EIS was used was considered as a service evaluation and therefore a separate ethical approval was not required as per consultation with Helsinki-Uusimaa Hospital District Ethical Committee. For the historical reference cohort an ethical approval was received from Helsinki-Uusimaa Hospital District Ethical Committee (ref. no. 130/13/03/03/2013). The consent for participation was obtained for women in the reference cohort. In the EIS cohort the consent for participation was not required since the study was considered as a service evaluation as per consultation with Helsinki-Uusimaa Hospital District Ethical Committee.

**Provenance and peer review** Not commissioned; externally peer reviewed.

**Data availability statement** Data are available upon reasonable request.

**ORCID iD**
Laura Bergqvist http://orcid.org/0000-0002-2047-9392

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
