## [Reviewer comments · BMJ Open]

ARTICLE DETAILS

TITLE (PROVISIONAL)	Added value of electrical impedance spectroscopy in adjunction of colposcopy: a prospective cohort study
AUTHORS	Bergqvist, Laura; Heinonen, Annu; Carcopino, Xavier; Redman, Charles; Aro, Karoliina; Kiviharju, Mari; Virtanen, Seppo; Omar, Pirjo-Liisa; Kotaniemi-Talonen, Laura; Louvanto, Karolina; Nieminen, Pekka; Kalliala, Ilkka

VERSION 1 – REVIEW

REVIEWER	Vavoulidis , Eleftherios Aristotle University of Thessaloniki, 2nd Dpt of Obstetrics & Gynecology
REVIEW RETURNED	09-Jun-2023

GENERAL COMMENTS	Excellent work. I have 4 points for futher investigation on your behalf (page 7, page 15, page 17 and page 25 of the submitted pdf file). Overall, the manuscript is very well-written with organized content, very good use of English language and statistics. (The reviewer provided a marked copy with additional comments. Please contact the publisher for full details.)
---

REVIEWER	Oh, Tong In Kyung Hee University
REVIEW RETURNED	17-Jun-2023

GENERAL COMMENTS	This is the cohort study to present the influence of EIS with Colposcopy for diagnosis of CIN. The authors presented EIS combined with colposcopy increased the diagnostic testing accuracy of CIN2+ compared to conventional colposcopy in women referred to colposcopy for abnormal cervical cytology. Reducing the number of biopsies and improving the performance of diagnosis for CIN are important and meaningful for studying. I would like to ask for some amendments. Comments: (1) Are there any changes in the demographic characteristics between the 2013-2017 reference cohort and the 2018-2021 study cohort? For example, a lower age of onset or a lower incidence rate? In the current version of Table 1, it is impossible to know other characteristic information, such as the age for CIN incidence, menopause, or history of gravida of patients with the disease. You need to look at various features that can generate differences in results between the two cohorts.
---

	(2) Is it enough to measure 10 to 12 points clockwise around the cervix using ZedScan for diagnosis of CIN? If you could compare the EIS result at each measuring point and the cytological results of the corresponding point, it would be a direct indicator of the adjunctive function of EIS for improving Colposcopy's performance. (3) If the ability to indicate the most abnormal cervical tissue area was excellent when using ZedScan, was the detection rate high in the acquired samples? How can you evaluate this? (4) Reducing the number of biopsies and maintaining diagnostic performance is very meaningful. However, does the average number of biopsy points between the two cohorts reflect the trends of different times? Recently, we wanted to try to reduce the number of biopsy points. (5) Reducing the number of biopsies is ultimately a sampling issue, so how about applying frequency difference electrical impedance imaging for the entire cervical region to solve this problem? (6) A fundamental problem of colposcopy is that it is highly dependent on the ability of the examiner to find suspected areas of CIN and perform a biopsy. How can you correct the difference in the ability of examiners between the two cohorts? Can the use of ZedScan compensate for examiners' dependence? What is the rationale for that? (7) In the results of this study, the biggest issue when ZedScan was used as an adjunctive device for Colposcopy was that its sensitivity was higher than that of the reference cohort, but its specificity was low. Should this be understood to require more samples still needed? (8) Why did you miss 11 high-grade lesions? Is there a sampling issue? Or Limitations of the current method? Or other reasons? (9) What is the probability of an error due to adenocarcinoma? (10) As you mentioned, "The increased detection of CIN2+ cases by EIS has been reported as most pronounced in women with low-grade cytology or with high-risk HPV positivity without cytological changes." I agree with you, and have there been cases like this among the data you have obtained? If there was, I think it would be very meaningful if only the corresponding samples were separated and analyzed. Minor comments: (11) ZedScan (company, Province, Country)
--	--

VERSION 1 – AUTHOR RESPONSE

Reviewer: 1

Dr. Eleftherios Vavoulidis , Aristotle University of Thessaloniki

Comments to the Author:

*** Please find additional comments from this reviewer in the attached file ***

Excellent work.

I have 4 points for further investigation on your behalf (page 7, page 15, page 17 and page 25 of the submitted pdf file).

Overall, the manuscript is very well-written with organized content, very good use of English language and statistics.

1. What about women with previous history of CIN treatment and possible residual disease? Where there such cases and if yes, where they included or excluded from the study?

Thank you. Women with previous history of CIN treatment were excluded from the study. We have now added this information in the methods section on page 5, row 112-113. The sentence now reads as:

“Exclusion criteria were transformation zone type 3 (TZ3), previous history of cervical cancer or large loop excision of the transformation zone (LLETZ) and pregnancy.”

This information has also been added to the flow chart, Figure S1 with corrected spelling of the word cervical (figure S1).

2. There are some later works about ZedScan and Colposcopy such as Tsampazis et al 2023 Macdonald et al 2017, Booth et al 2019, Booth et al 2020.

Possibly including them in your discussion if relate since they are not from the group of Tidy & Brown that basically invented the device??

Thank you for pointing this out. We have added Tsampazis et al. 2023 and Macdonald et al. 2020. The papers by Booth et al. studied another adjunctive technology, Dysis, instead of Zedscan. Their 2020 publication did, however, discuss the number of biopsies taken during colposcopy and we have referred to that paper accordingly.

Tsampazis et al 2023 has been added on, page 4, row 96, page 5, row 103 and page 13, row 261.

Macdonald et al 2020 has been added on, page 4, row 98.

Booth et al 2020 has been added on, page 14, row 283-284 along with the text referring to the paper: “A Danish study found taking four biopsies to increase the detection rate of cervical dysplasia to 95.2%.”

3. Were the same colposcopists involved in both groups or not? If there were different doctors involved in the colposcopies of these two groups then some potential bias may be considered due to different level of diagnostic skills and objectivity. Could this possibly explain the difference in average biopsies taken between the two cohorts?

Thank you for pointing out this important question. In both cohorts the colposcopists were experienced and certified by the Finnish Colposcopy Society.

EIS is a non-visual technique that measures epithelial changes in the impedance to the flow of the electric current. Consequently, when using ZedScan as an adjunct to conventional colposcopy the ability of the examiner to find suspected area is less important. ZedScan has been found to be beneficial particularly in colposcopies with minimal visual findings. When visual lesions are minimal or absent, it is the guidance of ZedScan which helps colposcopist to find the most suspected area for biopsy not the ability of colposcopist.

We do, however, agree that EIS device is not truly independent of colposcopic skills and the colposcopic performance can vary depending on the colposcopist, which again can affect diagnostic testing accuracy of colposcopy, with or without EIS. Furthermore, the referral cytology and colposcopic impression are incorporated in the analysis of ZedScan. We have now made an extension to the discussion section.

In order to take into account the variation of colposcopic performance and reliance on EIS device, we used large reference cohort representing routine work. In our opinion having several colposcopist performing the colposcopic examinations should be considered as a strength here, i.e. reflecting the real-life performance of a given intervention (here EIS), compared to studies with only one clinician or

colposcopist performing all procedures. We have now added also a statement about this to the section of strengths and limitations.

We further analysed the average number of biopsies by cytology. When we restricted the analysis only to colposcopists who had performed colposcopies in both cohorts we found a clear difference between the cohorts. The average number of biopsies were constantly higher in the reference cohort compared to the EIS cohort. In other words, the colposcopist who performed colposcopies in both cohorts changed their manner to take biopsies when ZedScan was used as an adjunct technology. We have added a new table of average number of biopsies by cytology as supplement, table S2.

Table S2. Average number of biopsies by cytology in the electrical impedance spectroscopy (EIS) cohort and in the reference cohort, including only the colposcopists who performed colposcopies in both cohorts.

EIS cohort Reference cohort

Average number of biopsies

ASC-US 1.7 2.3

LSIL 1.8 2.2

ASC-H 2.0 2.7

HSIL 2.3 2.8

ASC-US: atypical squamous cells of undetermined significance; LSIL: low-grade squamous intraepithelial lesion; HSIL: high-grade squamous intraepithelial lesion; ASC-H: atypical squamous cells that cannot exclude HSIL

We have now added the following sentences in the section of strength and limitations page 15, rows 313-320, now read as

“It is not possible to rule out that there would not have been any variation in sensitivity or specificity between the cohorts in different time periods. EIS device is not truly independent of colposcopic skills and the colposcopic performance can vary depending on the colposcopist. Also, the referral cytology and the colposcopic impression are incorporated in the EIS analysis by ZedScan. In order to take into account the variation of colposcopic performance and reliance on EIS device we collected a large cohort representing routine work. Including colposcopic examinations by several different colposcopists represents real-life situation which could be considered as a strength compared to studies where colposcopies have been performed by a single colposcopist.”

Section of results, page 11, rows 237-238 and page 13, rows 245-246.

“Among colposcopists who performed colposcopies in both cohorts, the average number of biopsies by cytology were higher in all cytology groups in the reference cohort compared to EIS cohort. The average number of biopsies varied between 1.7-2.3 in the EIS cohort and between 2.2-2.8 in the reference cohort (Table S2).”

Section of discussion, page 14, rows 277-280.

” In addition, the average number of biopsies by cytology among colposcopists who performed colposcopies in both cohorts were constantly higher in the reference cohort compared to the EIS cohort reflecting a change in manner/threshold to take biopsies when ZedScan was used as an adjunct technology.”

Reviewer: 2

Dr. Tong In Oh, Kyung Hee University

Comments to the Author:

This is the cohort study to present the influence of EIS with Colposcopy for diagnosis of CIN. The authors presented EIS combined with colposcopy increased the diagnostic testing accuracy of CIN2+

compared to conventional colposcopy in women referred to colposcopy for abnormal cervical cytology. Reducing the number of biopsies and improving the performance of diagnosis for CIN are important and meaningful for studying. I would like to ask for some amendments.

Comments:

(1) Are there any changes in the demographic characteristics between the 2013-2017 reference cohort and the 2018-2021 study cohort? For example, a lower age of onset or a lower incidence rate? In the current version of Table 1, it is impossible to know other characteristic information, such as the age for CIN incidence, menopause, or history of gravida of patients with the disease. You need to look at various features that can generate differences in results between the two cohorts.

Thank you for this question. There have been no changes in the catchment area of women referred for colposcopy to Helsinki University colposcopy clinic. Also, the screening test and target ages, classification of cervical cytology (Bethesda), and the national referral criteria to colposcopy remained similar for all patients referred to colposcopy within the study period.

We do agree that there might be difference in background variable distribution between the cohorts. Of the variables mentioned, we only have age available for both cohorts.

We have now compared the age distribution by cytology between the two cohorts as suggested and added rows in table1 to describe these data. When different cytologies were stratified by age, the cohorts differed somewhat within those referred for ASC-US cytology. Overall, over half of the women (55.3%) in the EIS cohort were aged 30 to 44 years whereas in the reference cohort 43.4% women were younger than 30 years of age.

The main results by cytology in three different age groups are already presented in table S1, rows 11-13, 17-19, 23-25, 29-31.

“There was no obvious impact of age on specificity or sensitivity within different cytologies (Table S1).”

We have now added this information in the results section, page11 rows 222-223.

We have also now added rows to table 1. page 9, rows 15-40, now reads as

Referral cervical cytology stratified by age

ASC-US

<30 y 28 4.3 43 4.5

30-44 y 52 8.0 28 2.9

≥45 y 14 2.2 28 2.9

LSIL

<30 y 39 6.0 79 8.2

30-44 y 153 23.6 224 23.3

≥45 y 44 6.8 78 8.1

ASC-H

<30 y 72 11.1 90 9.4

30-44 y 90 13.9 120 12.5

≥45 y 30 4.6 27 2.8

HSIL

<30 y 31 4.8 75 7.8

30-44 y 54 8.3 102 10.6

≥45 y 9 1.4 23 2.4

AGC-NOS

<30 y 5 0.8 5 0.5

30-44 y 15 2.3 12 1.2

≥45 y 8 1.2 11 1.1

AGC-FN

<30 y 0 0.0 3 0.3

30-44 y 2 0.3 9 0.9

≥45 y 1 0.2 5 0.5

647 100.0 962 100.0

(2) Is it enough to measure 10 to 12 points clockwise around the cervix using ZedScan for diagnosis of CIN? If you could compare the EIS result at each measuring point and the cytological results of the corresponding point, it would be a direct indicator of the adjunctive function of EIS for improving Colposcopy's performance.

Thank you. According to Zilico (the manufacturer of ZedScan) measurements from 10 to 12 points around the cervix is sufficient (www.zilico.com / Tidy J, Brown B, Healey T, Daayana S, Martin M, Prendiville W, Kitchener H. Accuracy of detection of high-grade cervical intraepithelial neoplasia using electrical impedance spectroscopy with colposcopy. BJOG 2013;120:400–411.)

After routine measurements (10-12 around the cervix) in case of suspected presence of CIN2+ by ZedScan a particular single point mode can be used to localise more carefully the most abnormal area to be biopsied. In other words, the tip of the device is put on the cervix and the device shows red light in case of CIN2+ whereas the light is green if the tip/device is not in the right place and the threshold value for a biopsy is not exceeded.

We have now added this information in method section of the manuscript to specify the use of the device, page 6 rows 141-143 now reads as:

“After routine measurements (10-12 around the cervix) in case of suspected presence of CIN2+ by ZedScan, a particular single point mode can be used to localise more carefully the most abnormal area to be biopsied.”

(3) If the ability to indicate the most abnormal cervical tissue area was excellent when using ZedScan, was the detection rate high in the acquired samples? How can you evaluate this?

Thank you. We have evaluated this in results section in table 2, page 10, rows 8-13 and row 22. Histologically confirmed CIN2+ was found overall in 34.3% of women in EIS the cohort. In women with HSIL referral cytology 81.9% had CIN2+ whereas in women low-grade cytology the CIN2+ was detected among 16.2%.

Within the EIS cohort, Zedscan indicated CIN2+ lesions in 92.8% histologically confirmed CIN2+ cases while colposcopy alone within the same cohort had colposcopic impression of CIN2+ in 74.8% of detected CIN2+ cases. These results are already presented in page 11, rows 230-233.

However, in cases of suspicion of CIN2+ by ZedScan it is possible that the biopsies were not taken at the right place or that the result of histopathology was not right. Both cases can happen during colposcopy with or without EIS-examination. Consequently, it could be assumed that such cases would be equally common in both cohorts since the setting was similar and should therefore not explain possible differences between cohorts.

We do agree that longer follow-up time would have been needed in order to find out in greater detail false positive and false negative rates. We have already discussed this matter in page 16 rows 322-323 and 325-326.

“CIN2+ lesions could well have been missed in both cohorts since the results are based on data collected on the initial visit. However, complete certainty of the histology would have required LLETZ for all participants which would not have been ethically just.”

In case more detailed discussion on this is warranted, we are naturally happy to comply.

(4) Reducing the number of biopsies and maintaining diagnostic performance is very meaningful. However, does the average number of biopsy points between the two cohorts reflect the trends of different times? Recently, we wanted to try to reduce the number of biopsy points.

Thank you for pointing this out. In our colposcopy clinic, there has not been a tendency/requirement or trend to reduce the number of biopsies. According to Finnish current care guidelines, the threshold to take a biopsy is colposcopic impression of CIN1+, page 14, rows 270-272. Furthermore, in large lesions colposcopist may want to take several biopsies in order not to miss or misclassify a lesion. Otherwise, no clear time-dependent intervention on reducing biopsies has been implemented in the clinic.

We further analysed the average number of biopsies by cytology. When we restricted the analysis only to colposcopists who had performed colposcopies in both cohorts we found a clear difference between the cohorts. The average number of biopsies were constantly higher in the reference cohort compared to the EIS cohort. In other words, the colposcopist who performed colposcopies in both cohorts changed their manner to take biopsies when ZedScan was used as an adjunct technology. We have added a new table of average number of biopsies by cytology as a supplement, table S2

Table S2. Average number of biopsies by cytology in the electrical impedance spectroscopy (EIS) cohort and in the reference cohort, including only the colposcopists who performed colposcopies in both cohorts.

EIS cohort Reference cohort

Average number of biopsies

ASC-US 1.7 2.3

LSIL 1.8 2.2

ASC-H 2.0 2.7

HSIL 2.3 2.8

ASC-US: atypical squamous cells of undetermined significance; LSIL: low-grade squamous intraepithelial lesion; HSIL: high-grade squamous intraepithelial lesion; ASC-H: atypical squamous cells that cannot exclude HSIL

We have amended the text in the results section, page 11, rows 237-238 and page 13 rows 245-246, now read as

“Among colposcopists who performed colposcopies in both cohorts, the average number of biopsies by cytology were higher in all cytology groups (ASC-US, LSIL, ASC-H, HSIL) in the reference cohort compared to EIS cohort. The average number of biopsies varied between 1.7-2.3 in the EIS cohort and between 2.2-2.8 in the reference cohort (Table S2).

We have also added the following in the discussion section, page 14, rows 277-280, now read as
”In addition, the average number of biopsies by cytology among colposcopists who performed colposcopies in both cohorts were constantly higher in the reference cohort compared to the EIS cohort reflecting a change in manner/threshold to take biopsies when ZedScan was used as an adjunct technology.

Please also see our reply to a similar question from Reviewer1.

(5) Reducing the number of biopsies is ultimately a sampling issue, so how about applying frequency difference electrical impedance imaging for the entire cervical region to solve this problem?

Thank you for this comment. In principle, the 10 to 12 measurements around the cervix covers well the transformation zone in most women. However, we agree that it is possible that in certain cases the measurements may omit some minor areas of the cervix, especially when the cervix is very large. We have now added a sentence on this in methods, page 6, rows 139-141.

"In most women, 12 measurements cover well the junction area of the cervix. However, it might be possible that minor areas are omitted in case of very large cervix."

(6) A fundamental problem of colposcopy is that it is highly dependent on the ability of the examiner to find suspected areas of CIN and perform a biopsy. How can you correct the difference in the ability of examiners between the two cohorts? Can the use of ZedScan compensate for examiners' dependence? What is the rationale for that?

Thank you for pointing out this important question. In both cohorts the colposcopists were experienced and certified by Finnish Colposcopy Society. However, it is not possible to rule out that there would not have been any variation in sensitivity or specificity of individual colposcopists between the cohorts and in different time periods. We have therefore now extended the discussion section to cover this (changes made detailed below).

EIS is a non-visual technique to measure epithelial changes in the impedance to the flow of the electric current. Consequently, when using ZedScan as an adjunct to conventional colposcopy the ability of the examiner to find suspected area is less important. ZedScan has found to be beneficial particularly in colposcopies with minimal visual findings. When visual lesions are minimal or absent, it is the guidance of ZedScan which helps colposcopist to find the most suspected area for biopsy not the ability of colposcopist. We do agree that EIS device is not truly independent of colposcopic skills and the colposcopic performance can vary depending on the colposcopist. The referral cytology and colposcopic impression are therefore incorporated in the analysis of ZedScan. We have also made an extension to the discussion section (detailed below).

In order to take into account the variation of colposcopic performance and reliance on EIS device we used large reference cohort representing routine work. In our opinion, having several colposcopist performing the colposcopic examinations should be considered as a strength here, i.e. reflecting the real-life performance of a given intervention, compared to studies with only one colposcopist. We have now therefore added the following to the section of strengths and limitations.

Page 15, rows 313-320, now read as:

"It is not possible to rule out that there would not have been any variation in sensitivity or specificity between the cohorts in different time periods. EIS device is not truly independent of colposcopic skills and the colposcopic performance can vary depending on the colposcopist. Also, the referral cytology and the colposcopic impression are incorporated in the analysis by ZedScan. In order to take into account the variation of colposcopic performance and reliance on EIS device we collected a large cohort representing routine work. Including colposcopic examinations by several different colposcopists represents a real-life situation which could be considered as a strength compared to studies where colposcopies have been performed by a single colposcopist."

Please also see our reply to a similar question from Reviewer1.

(7) In the results of this study, the biggest issue when ZedScan was used as an adjunctive device for Colposcopy was that its sensitivity was higher than that of the reference cohort, but its specificity was low. Should this be understood to require more samples still needed?

Thank you. It is possible that biopsies were not taken at optimal area which would lead to the requirement to take more biopsies. Another possibility is that ZedScan indicated the presence of CIN2+ when such lesion was not present especially in women with ASC-H referral cytology. In women with ASC-H referral cytology the prevalence of CIN2+ was much lower than after HSIL referral cytology, possibly related to the differences in cytological classifications between countries as already presented in page 15, rows 296-299.

Moreover, the threshold for biopsy could have differed slightly between the cohorts. At least biopsies with colposcopic impression of LSIL+ could have been more common in the reference cohort than in the EIS cohort. We have discussed this in page 14, rows 270-277.

“One explanation for lower prevalence of CIN2+ lesions in the EIS cohort after LSIL and ASC-H cytology could be that routine practice in Finland is to take biopsies also from low-grade lesions, rather than to abstain from taking biopsies when CIN2+ lesions are not colposcopically suspected. Biopsies even from mild acetowhite lesions are important in excluding a high-grade disease as the sensitivity of colposcopy to detect CIN2+ is far from 100%. Such biopsies could well have been more frequent without than with EIS as an additional confirmation on suspected absence of CIN2+. This is supported by the observation that two or more biopsies were taken from 54% of women in the EIS cohort, whereas up to 75% of women in the reference cohort had at least two biopsies.”

In case further discussion on this is warranted, we are naturally happy to comply.

(8) Why did you miss 11 high-grade lesions? Is there a sampling issue? Or Limitations of the current method? Or other reasons?

Thank you. The colposcopic impression was normal in both AIS cases and Zedscan did not indicate presence of CIN2+ either. We can only speculate why, perhaps the lesions were not visible and situated in deeper epithelial layers. Sensitivity of colposcopy alone, and even with EIS as an adjunctive technology, is not 100%.

According to Finnish current care guidelines the threshold for biopsy is colposcopic impression of LSIL or worse, which usually leads to higher number of biopsies and therefore lower sensitivity, but in these data it might also have led to higher CIN2+ prevalence overall, page 14, row 271-272.

In other histopathological HSIL cases the colposcopic impression was low-grade in nine cases and normal in two cases and the decision to take a biopsy was made by the colposcopist because Zedscan did not alarm.

ZedScan performs the analysis of the area under the tip of the snout. If the lesion is deeper in the cervical channel and not visible or accessible it is impossible for the device to recognise it. We agree that ZedScan has its limitations, sensitivity and specificity of this method are not 100%. ZedScan might miss some lesions that either could have been detected with lower biopsy threshold or where biopsy would not have been indicated even in conventional colposcopy.

We have already discussed this matter in page 14, rows 290-291.

“If lesions were missed, it could possibly be due to a higher biopsy threshold in the EIS cohort, as indicated by lower number of biopsies.”

And we have now added the following to strengths and limitations, page 16 row 323-325.

“EIS might miss some lesions that either could have been detected with lower biopsy threshold or where biopsy would not have been indicated even in conventional colposcopy.”

(9) What is the probability of an error due to adenocarcinoma?

Thank you. It is impossible us to comment on the probability of ZedScan to detect or miss an adenocarcinoma. ZedScan should differentiate between normal, pre-cancerous and cancerous tissue. Tidy et al. (EJGO 2018) reported that “changes in the glandular epithelium associated with HG- CGIN demonstrate similar changes to EUS to those of CIN.”

Moreover, Pathiraja et al. (2020) reported in a systematic review that “the majority of the squamous cell carcinomatous tissue seemed to result in significantly lower impedance spectra, in contrast with the adenocarcinomatous tissue studies, which gave a more variable spread of impedance changes. This finding was consistently seen throughout all the studies of cervical, skin and oral squamous cell carcinomas (SCC), where the SCC tissue gave significantly lower SCC readings than its corresponding normal tissue.”

(10) As you mentioned, "The increased detection of CIN2+ cases by EIS has been reported as most pronounced in women with low-grade cytology or with high-risk HPV positivity without cytological changes." I agree with you, and have there been cases like this among the data you have obtained? If

there was, I think it would be very meaningful if only the corresponding samples were separated and analyzed.

Thank you for this question. We agree that the performance of Zedscan should be analysed stratified according to referral smear, as the prevalence of endpoint, CIN2+ lesions, depends on referral cytology. All results have therefore already been presented stratified according to referral cytology. (Page 10, table 2, rows 8-12, 22).

Women with persistent HPV positivity without cytological changes were excluded from the study because of insufficient number of women in the reference cohort.

Minor comments:

(11) ZedScan (company, Province, Country)

Thank you. Here is the information on ZedScan's manufacturer. Zilico Ltd, Manchester, United Kingdom. Available: <https://zilico.co.uk>

VERSION 2 – REVIEW

REVIEWER	Vavoulidis , Eleftherios Aristotle University of Thessaloniki, 2nd Dpt of Obstetrics & Gynecology
REVIEW RETURNED	21-Aug-2023
GENERAL COMMENTS	This revised version of the manuscript is way better than the initial one with better reading flow and in-depth analysis (when required).

VERSION 2 – AUTHOR RESPONSE